# A Missense Variant in CASKIN1’s Proline-Rich Region Segregates with Psychosis in a Three-Generation Family

**DOI:** 10.3390/genes14010177

**Published:** 2023-01-09

**Authors:** Marah H. Wahbeh, Xi Peng, Sofia Bacharaki, Alexandros Hatzimanolis, Stefanos Dimitrakopoulos, Elizabeth Wohler, Xue Yang, Christian Yovo, Brady J. Maher, Nara Sobreira, Nikos C. Stefanis, Dimitrios Avramopoulos

**Affiliations:** 1McKusick-Nathans Department of Genetic Medicine, Johns Hopkins School of Medicine, Baltimore, MD 21205, USA; 2Predoctoral Training Program in Human Genetics and Molecular Biology, Johns Hopkins School of Medicine, Baltimore, MD 21201, USA; 3Department of Psychiatry, General Hospital of Syros, 84100 Cyclades, Greece; 4Department of Psychiatry, School of Medicine, National and Kapodistrian University of Athens, Eginition Hospital, 15772 Athens, Greece; 5Lieber Institute for Brain Development, Baltimore, MD 21205, USA; 6The Solomon H. Snyder Department of Neuroscience, Johns Hopkins School of Medicine, Baltimore, MD 21205, USA; 7Department of Psychiatry and Behavioral Sciences, Johns Hopkins School of Medicine, Baltimore, MD 21205, USA

**Keywords:** iPSC, stem cells, CRISPR/Cas9, schizophrenia, bipolar disorder, psychiatric disease, psychosis, CASKIN1, CASK, NRXN1

## Abstract

The polygenic nature of schizophrenia (SCZ) implicates many variants in disease development. Rare variants of high penetrance have been shown to contribute to the disease prevalence. Whole-exome sequencing of a large three-generation family with SCZ and bipolar disorder identified a single segregating novel, rare, non-synonymous variant in the gene *CASKIN1*. The variant D1204N is absent from all databases, and *CASKIN1* has a gnomAD missense score Z = 1.79 and pLI = 1, indicating its strong intolerance to variation. We find that introducing variants in the proline-rich region where the D1204N resides results in significant cellular changes in iPSC-derived neurons, consistent with *CASKIN1*’s known functions. We observe significant transcriptomic changes in 368 genes (padj < 0.05) involved in neuronal differentiation and nervous system development. We also observed nominally significant changes in the frequency of action potentials during differentiation, where the speed at which the edited and unedited cells reach the same level of activity differs. Our results suggest that *CASKIN1* is an excellent gene candidate for psychosis development with high penetrance in this family.

## 1. Introduction

Like many neuropsychiatric disorders, schizophrenia (SCZ) is a common, severe, and highly heritable disorder with a complex etiology. It is associated with an increased risk of depression [1] and suicide [1,2]. Together with its devastating cognitive symptoms, SCZ is a major public health burden.

The genetic architecture of SCZ is polygenic, meaning that many variants, common and rare, have been implicated in disease development [3,4]. Genome-wide association studies (GWAS) have identified a large number of common variants to be associated with SCZ, the most recent one reporting 287 associated loci [3]. Most of these loci are in non-coding genomic regions or concentrated in genes that are expressed in excitatory and inhibitory neurons. Functional validation of these variants can be difficult to model because of their small effect size. Family studies [5], large-scale case-control studies [6], whole-genome sequencing studies (WGS) [7] and exome sequencing studies [4] over the last few decades have identified rare variants with high penetrance that have a higher associated risk with SCZ compared to GWAS-identified common variants. The majority of the currently known rare variants are copy number variants (CNVs), although single nucleotide variants (SNVs), notably mutations in *SETD1A* [8] and *RBM12* [9], have also been implicated. SCZ’s association with CNVs has been extensively studied [10,11,12], as some confer the greatest associated risk to disease. These CNVs include duplications at 16p11.2, 1q21.1, 7q11.23 (Williams-Bueren Syndrome), 15q11-q13 (Angelman Syndrome/Prader-Willi Syndrome) and deletions in 1q21.1, 22q11.2, 3q29, 15q13.3, and 2p16.3 [12,13,14,15]. All but one of these CNVs impact multiple genes, some affecting 20. Deletions in 2p16.3 are the only known SCZ-associated CNVs that impact a single gene, *NRXN1*. Individuals with *NRXN1* CNVs have ~ a 10-fold increase in risk of developing SCZ [12,16,17].

*NRXN1* is a member of a family of proteins called neurexins. Neurexins are transmembrane proteins that primarily reside in the pre-synaptic terminal of neurons [18]. They interact with neuroligins, which are postsynaptic adhesion molecules and function together as cell-adhesion molecules connecting neurons, mediating signaling, and specifying synaptic functions [18,19]. Outside the cell membrane, *NRXN1* interacts with neuroligins from the corresponding dendritic spine while its cytoplasmic tail interacts with competing tri-partite protein complexes of CASK/VELIS (gene names *CASK/LIN7A)* and CASKIN1 (gene name *CASKIN1*) or MINT1 (gene name *APBA1)* [20]. In this manner, *NRXN1* is involved in synapse formation, maintaining synaptic connections, and neurotransmission, regulated in part by the competition of *MINT1*-*CASKIN1*. Moreover, within this pathway, *CASK* has been associated with autism spectrum disorder (ASD), and other neuro developmental disorders, such as microcephaly with pontine and cerebellar hypoplasia (MICPCH), and X-linked intellectual disability (XLID) [21,22,23]. Here, we report a large three-generation family segregating psychosis as an autosomal dominant trait with high penetrance, with a co-segregating previously unknown coding variant in *CASKIN1*. Exome sequencing revealed no other segregating variant, and the only two non-penetrant individuals in the family have the lowest polygenic risk scores (PRS) for SCZ. CRISPR based introduction of the variant in induced pluripotent stem cells (iPSCs) and subsequent differentiation to excitatory glutamatergic neurons supports that the region the variant occupies is functionally important, with 368 genes significantly changing expression compared to otherwise isogenic cells. The edited neurons also showed significant differences in activity, as measured by calcium imaging. Taken together with *CASKIN1*’s close relationship with *NRXN1*, our data suggest that the identified variant is a strong candidate to be causing psychosis with high penetrance in this family.

## 2. Materials and Methods

### 2.1. Subjects 

Subjects were recruited from SD and SB. All subjects are part of the same pedigree, exclusively of Greek origin. Clinical evaluation for each subject was based on M.I.N.I. (Mini International Neuropsychiatric Interview) and supplemented by a detailed review of medical records for all affected individuals. Final best estimate diagnoses were based on DSM-IV criteria after a 2-stage consensus diagnosis (SD, SB and NSt). The study protocol was approved by the institutional ethical committee and all the participants provided written informed consent in both Greek and English. The subjects did not consent to public data sharing through dbGAP or other databases.

### 2.2. Exome Sequencing

Exome sequencing was performed at the Baylor-Hopkins Center for Mendelian Genomics, utilizing the Agilent SureSelect HumanAllExonV5Clinical_S06588914 for library preparation. Libraries were sequenced on the HiSeq2500 platform using 125 bp paired end runs and sequencing chemistry kits HiSeq Rapid PE Cluster Kit v4 and HiSeq SBS Kit v4. Fastq files were aligned with BWA [24] version 0.7.8 to the 1000 genomes phase 2 GRCh37/hg19 human genome reference. Duplicate molecules were flagged with Picard version 1.74. Local realignment around indels and base call quality score recalibration was performed using the Genome Analysis Toolkit (GATK) [25] version 3.0-0. Variant filtering was conducted using the Variant Quality Score Recalibration (VQSR) method [26]. Analysis of variant call files (vcf) were performed using the PhenoDB analysis tool [27] to identify rare (MAF < 0.001 in the 1000 Genomes project, Exome Variant server, gnomAD, and our in-house database), functional variants (missense, nonsense, frameshift, splicing, or stoploss) with recessive or dominant inheritance patterns. The MAF was chosen to be compatible with the predicted effect size of our variant (~80 given the predicted 80% penetrance and a ~1% prevalence of SCZ). Higher frequency alleles with this effect size would have been easily detected by large-scale sequencing studies for psychotic disorders.

### 2.3. Variant Selection and Genotyping 

The variants fulfilling the criteria above were further genotyped by PCR and Sanger sequencing using primers (25 nmole standard desalted, IDT) on all family members with available DNA. Genotyping of iPSC cell line used for CRISPR/cas9 editing and for off target analysis was conducted by PCR and Sanger sequencing using the primers listed in Appendix A.

### 2.4. Linkage Analysis

Parametric linkage analysis was performed using the program merlin [28] using all genotyped individuals and the newly identified variant as the sole marker. For linkage purposes all individuals with a psychiatric diagnosis were considered affected and all examined individuals without a diagnosis were considered unaffected. The following linkage parameters were applied: Disease allele frequency 10^−9^, heterozygote penetrance 0.80, homozygote penetrance 1.0, phenocopy frequency 0.01 (the prevalence of SCZ).

### 2.5. Genome-Wide Genotyping, Quality Control, and Authentication of iPSC Line

Genome-wide genotyping of all members of the family with available DNA performed using the Illumina (Illumina Inc, San Diego, CA, USA) platform using the Multi-Ethnic Global Diversity Array at The Johns Hopkins Genetics Core Facility. This included a total of 1,711,632 with a mean call rate of 0.9979. Genotypes were used for the generation of PRS, to test the integrity of the cell lines and to authenticate them.

To confirm the identity of the iPSC line, Illumina Global Diversity Array-24 v2.0 was performed at the Johns Hopkins University School of Medicine genetics core resource facility on the line used before editing and on the clones after editing. This excluded any major chromosomal abnormalities—deletions or duplications greater than 1 Mb. A minor chromosomal abnormality was found in 3 edited and 1 unedited clone, of which 1 edited and 1 unedited were among those used for Ca^2+^ imaging. The abnormality was a mosaic gain at location chr4: 129,218,126–130,051,748 (GRCh37/hg19), an 800 kb region with 4 genes. However, these genes also show multiple polymorphic copy number gains in the Database of Genomic Variants that are not associated with phenotypes (e.g., essv11597514 and nssv3639434). Further, none of these genes were found differentially expressed between the edited and unedited neurons, which we considered an additional indication that it is unlikely to be a cause of concern. We performed an Identity by State (IBS) analysis of the SNPs from the array using plink (http://pngu.mgh.harvard.edu/purcell/plink/ accessed on 10 January 2021) [29] and found that the clones were 99.979–100% identical to each other and 99.9409–100% identical to the original line before editing, all within the array error rate.

### 2.6. Polygenic Risk Score (PRS) Calculation

PRSs were calculated using the genome-wide genotyping results for all 8 carriers of the novel variant allele with the PRSice package [30] and the PGC-2 summary data (https://pgc.unc.edu/) as a reference GWAS (Appendix A). Phenotypes for *p*-value threshold testing were assigned as penetrant (*n* = 6) or non-penetrant (*n* = 2). To assess the significance of the result, scores were recalculated for all possible distributions of phenotypes (8 choose 2 = 28).

### 2.7. Cell Line Used for CRISPR Editing

The iPSC line (MH0180966) used for CRISPR/cas9 editing and further experiments was derived from a female individual of European decent without a psychiatric diagnosis and provided to us by the stem cell repository at Rutgers University (NIMH/RUCDR). We received the line at passage number 12.

### 2.8. Cell Culture

iPSCs were cultured in StemFlex media supplemented with serum (Gibco #A3349401, Thermofisher Scientific, Waltham, MA, USA ) at 37 °C with 5% CO_2_ on laminin coated plates at 5 μg/mL (BioLamina #LN521, Thermofisher Scientific, Waltham, MA, USA) that were incubated at 37 °C for at least 2 h. Passaging was conducted by re-plating cells after detaching them at ~70–90% confluency using Accutase (MilliporeSigma #A6964, Burlington MA, USA) following a quick wash with 1X Phosphate-buffered saline (PBS). 10 μM ROCK inhibitor (Y-27632 dihydrochloride, Tocris #1254, Bristol UK) was added on the day of passaging, as well as thawing and transfection days. Cells were frozen in StemFlex media with 10% Dimethylsulfoxide (DMSO).

### 2.9. sgRNA and ssODN Design

sgRNAs were designed using the CRISPR design tool provided by Horizon (https://horizondiscovery.com/en/ordering-and-calculation-tools/crispr-dna-region-designer). 2 sgRNAs were selected both within 10 bp from the cut site to increase homology directed repair chances but only sgRNA 1 was used in further experiments (Appendix A). 160 bp ssODN repair template (sigma) was designed to have 4 bp changes to the original sequence. These included the variant change and 3 additional changes to increase HDR efficiency and allow for ease of screening. Sequences for the sgRNA and ssODN repair template is listed in Appendix A.

### 2.10. Cloning

Restriction enzyme BbsI sticky ends were added to single-stranded DNA oligos (25 nmoles standard desalted, IDT) of each strand of the designed sgRNAs (Appendix A). The single stranded oligos were annealed and cloned into the PX459 plasmid (Addgene #62988, Watertown MA, USA) as described by the Thomas lab [31]. 5–10 μL of the cloning mixture was transformed into 25–50 μL of One Shot TOP10 Chemically Competent *E. coli* (ThermoFisher #C404003, Waltham MA, USA) following the manufacturer’s protocol. Carbenicillin LB agar plates were used to plate the transformed bacteria on and incubated at 37 °C overnight. Separate bacterial colonies were picked and grown into 3–5 mL of LB broth with Carbenicillin and cultured at 37 °C overnight. Plasmid DNA from bacterial clones was isolated using Qiagen mini prep kit (Qiagen #27104, Hilden Germany). Successful cloning was confirmed by Sanger sequencing.

### 2.11. Transfection, Selection, and Screening

Cells were plated 24 h before transfection at 40 k cells/well on a 24-well laminin coated plate. Using 2 uL of Lipofectamine Stem (ThermoFisher #STEM00003, Waltham MA, USA) and following the manufacturer’s instructions, the cells were transfected with 500 and 250 ng of PX459 (with the sgRNA cloned in) and the ssODN repair template at 0.004 uM concentration. Cells were incubated in Lipofectamine mixture for 5 h instead of the recommended 4. Puromycin was added ~40 h post transfection at 1 ug/mL concentration and was removed for 24 h after. A proportion of the cells were harvested a week later and before passaging for bulk screening with PCR and restriction digestion. For screening, a proportion of the bulk cells from transfected wells were collected and extracted using Quick Extract buffer (Lucigen #QE0905T, Middleton WI, USA) following the protocol published by the manufacturer. PCR of the *CASKIN1* target site was conducted using 5 μL (QuickExtract, biosearchtech, Hoddesdon, UK) or 1 μL (Gentra Puregene, Minneapolis, MN, USA) of extracted DNA, primers flanking the target region (Appendix A) and AccuPrime Taq DNA Polymerase (ThermoFisher #12339016, Waltham MA, USA). 5 μL of the PCR product was combined with 10X Bromophenol Blue loading dye and loaded into 1–2% agarose gels and run at 80–120 V for 30–60 min. Once the amplicon was confirmed, 15 μL of the remaining PCR product was digested using 1 μL of Hha1 restriction enzyme (ThermoFisher #ER1851, Waltham MA, USA) for 2 h at 37 °C. Digested product was run on 2% agarose or 1%/3% agarose/NuSeiv gels at 100–120 V for 60–90 min.

The cells that were transfected with 500 ng of PX459 were single-cell plated on a 6-well plate at 300–500 cells per well density. Once the cells grew into colonies, they were manually picked. Half the clone went into a PCR tube and the other half went into one well of a 24-well culture plate The PCR tube was mixed with 10 μL of Quick Extract buffer to extract DNA, which was then followed by PCR, restriction digestion and Sanger sequencing to confirm genotype. All chosen clones (edited and unedited) were tested for off-target edits.

### 2.12. Off-Target Analysis

Off-target sites were identified using the CRISPOR tool (http://crispor.tefor.net/). We prioritized off target sites that were most likely to be targeted (had 3 or less mismatches with our sgRNA). PCR primers for each site were designed to flank the identified off-target sites (Appendix A). All clones were screened for edits at these sites by PCR, as described above, and Sanger sequencing.

### 2.13. Differentiation into Glutamatergic Neurons

Each iPSC clone was differentiated into excitatory neurons using the induced differentiation protocol described by Zhang et al. [32]. In brief, iPSCs were plated at 250 k/well of a 6-well plate and were transfected with Ngn2 (10 μL) and rTTA (10 μL) lentiviruses 24–72 h post seeding (at ~40% confluency) to achieve a multiplicity of infection (MOI) between 1 and 10. We used 1 μL of 1 μg/mL Polybrene per mL of media (Santa Cruz #SC-134220, Dallas TX) to facilitate the transduction. The media was changed at 4 h post transduction. The virus-infected cells were then expanded, and frozen stocks were made for future differentiation. Ngn2 and rTTA lentiviruses were obtained from the University of Pennsylvania Core store. 250 k cells of each transduced clone were plated on laminin coated 6-well plates (DIV-2). Ngn2 expression was induced with 2 μg/mL Doxycycline (Sigma #D9891) on DIV 0 using an induction media made of DMEM/F12 (ThermoFisher #11330-032, Waltham, MA, USA), 1X N2 (ThermoFisher #17502048, Waltham, MA, USA), 1.5 mg/mL D-Glucose (ThermoFisher #G8270-100G), 0.1% 2-bME (ThermoFisher 21985023, Waltham, MA, USA), 100 μg/mL Primocin (Invivogen ant-pm-1, San Diego, CA, USA), 10 ng/mL BDNF (Peprotech #AF-450-02, Rocky Hill, NJ, USA), 10 ng/mL NT3 (Peprotech #450-03, Rocky Hill, NJ, USA), 200 ng/mL Laminin (Millipore Sigma #L2020, Burlington, MA, USA) and 2 μg/mL Doxycycline (Sigma #D9891, St. Louis, MO, USA). Transduced cells were selected for using 2 μg/mL puromycin on DIV 1, 24 h post Doxycycline induction using the same induction media. Surviving cells were plated on 1:35 matrigel (Corning #354230, Corniing, NY, USA) coated 24-well plates on DIV 2 at 100 k cells/well in neural differentiation media made up of Neurobasal media (ThermoFisher #21103-04, Waltham, MA, USA), 1X B27 (ThermoFisher #17504044, Waltham MA, USA), 1X Glutamax (ThermoFisher #350500-61, Waltham, MA, USA), 1X Penn/Strep (ThermoFisher #15140-122, Waltham, MA, USA), 6 mg/mL D-Glucose (ThermoFisher #G8270-100G, Waltham MA, USA), 10 ng/mL BDNF (Peprotech #AF-450-02, Rocky Hill, NJ, USA), 10 ng/mL NT3 (Peprotech #450-03, Rocky Hill, NJ, USA), 200 ng/mL Laminin (Millipore Sigma #L2020) and 2 μg/mL Doxycycline (Sigma #D9891). Cells were fed with a 50% media exchange of neural maturation media made up of Neurobasal media A (ThermoFisher #A24775, Waltham, MA, USA), 1X B27 (ThermoFisher #17504044, Waltham, MA, USA), 1X Glutamax (ThermoFisher #350500-61, Waltham, MA, USA), 1X Penn/Strep, Glucose Pyruvate mix (1:100, final concentration of 5 mM glucose and 10 mM sodium pyruvate), 10 ng/mL BDNF (Peprotech #AF-450-02, Rocky Hill, NJ, SUA), 10 ng/mL NT3 (Peprotech #450-03, Rocky Hill, NJ, USA), 200 ng/mL Laminin (Millipore Sigma #L2020, St. Lois, MO, USA) and 2 μg/mL Doxycycline every other day until DIV 12. We add 2 μM of Cytosineb-D-arabinofuranoside hydrochloride (Ara-C) (Sigma #1162002-250MG, St. Louis, MO, USA) on DIV 4 to arrest proliferation and remove non-neuronal cells. Doxycycline induction was discontinued at day 12 and cells were fed with 50% media exchange every two days until DIV 21 with the Neural maturation media. Neurons were harvested for RNA-seq and further analysis on DIV 21.

### 2.14. RNA Isolation, Sequencing, and Data Analysis

Total RNA was isolated using Zymo RNA (#R1054) following the manufacturer’s protocol from 12–24 wells of a 100 k/well of neurons for each cell line. The RNA from the iPSC-derived neurons was submitted to Novogene (novogene.com) for 150 bp paired-end RNA-sequencing. All libraries passed Novogene’s post-sequencing quality control. We received on average 42.8 million reads per sample with a maximum of 46.3 and a minimum of 41.8 million. The reads were aligned to the human reference genome GRCh38 by Hisat2 version 2.1.0 [33]. Samtools 1.9 [34] produced corresponding BAM files. Stringtie 2.0.3 [35] was used to assemble and estimate the abundance of transcripts based on GRCh38 human gene annotations [36]. Bioconductor package tximport computed raw counts by reversing the coverage formula used by Stringtie with the input of read length [37]. The output was then imported to another Bioconductor package, DESeq2, for differential gene expression analysis [38]. Benjamini–Hochberg adjustment was used to calculate adjusted *p*-values (padj).

### 2.15. RNA-Seq Data Enrichment Analysis

The normalized gene expression levels calculated by DEseq were used for input to the GSEA analysis package. Gene sets were manually selected from the Hallmark gene sets (see http://www.gsea-msigdb.org/gsea/msigdb) to contain keywords such as nervous system, brain and glutamate. The DE genes were also uploaded to the STRING database [39] (string-db.org) to look for interactions within the set of genes, not including any additional level of interactors to ensure the validity of the observed network enrichment *p*-values as suggested by the curators of the web site. We used the highest-level confidence (0.7) for interactions.

### 2.16. Ca^2+^ Imaging

iPSCs were plated on a bed of rat astrocytes (kindly gifted from LIBD) on PDL-laminin (Millipore Sigma #L2020, St. Louis, MO, USA) coated plates (ibidi #NC1005660, Gräfelfing, Germany) at 150 K cells/well and differentiated as described above with some changes. The only major change was the replacement of Neurobasal media A (ThermoFisher #A24775 Waltham, MA, USA) with Neurobasal Plus media (ThermoFisher #A3582901 Waltham, MA, USA) to help with neuronal activity. At DIV 4, each well was infected with 1.2 μL of AAV1-hSyn1-mRuby2-P2A-GCaMP6s (Addgene #50942-AAV1, Watertown, MA, USA). Imaging was performed live in media at DIV 21, DIV 28 and DIV 35 on a Zeiss LSM780 (Zeiss, Oberkochen, Germany) with a 10×/0.45 NA objective and a temperature/atmospheric controlled environment (LIBD). An mRuby fluorescence reference image and a time-series of GFP were acquired at 4 Hz for 8 min for each well. Analysis was conducted using the CapTure pipeline [40]. T-test was used to calculate *p*-values listed in all figures.

## 3. Results

### 3.1. A Large Family Segregating Psychosis in an Apparent Autosomal Dominant Fashion

The proband (III-9), male, aged 27 at the time of diagnosis, presented with a manic episode and was given the diagnosis of bipolar disorder I (BPD-I) (Figure 1). He was treated with lithium and quetiapine. Up to today, he is being followed by SB and remains under lithium treatment. He had a hypomanic episode seven years later and has been symptom free since then. Family history revealed multiple cases of psychotic disorders in the family (Figure 1). Psychiatric evaluation of all available members of the family (see methods) identified four other members with SCZ and one with schizoaffective disorder (Figure 1, Appendix A). This revealed a three-generation pedigree with six affected members and an apparent autosomal dominant mode of inheritance (Figure 1). Incomplete penetrance appeared likely as individual (I-3) was an obligate carrier with no diagnosis of psychosis. However, this individual (I-3) had an anxiety disorder diagnosis. Note that only individuals shown in Appendix A have been examined.

### 3.2. Exome Sequencing Revealed Two Missense Variants in Genes Expressed in the Brain

We performed whole-exome sequencing (WES) through the Baylor-Hopkins Center for Mendelian Genomics on the proband (III-9) and three distant relatives with a phenotype (III-12, II-11, and II-13) (Figure 1). We filtered the identified variants for those present in all four affected individuals, having an allele frequency <0.1% in public databases and present in genes expressed in the brain. Only two variants met these filtering criteria, in the genes *RHEBL1* and *CASKIN1*. All individuals from the family with available DNA were then Sanger sequenced for both variants. This ruled out the variant in *RHEBL1* because, in addition to the affected, it was found in five unaffected individuals (I-2, I-3, II-3, II-5, III-10), one of which was in homozygosity (II-5). In contrast, the variant in *CASKIN1* perfectly segregated with psychosis (Figure 1) with the exception of two non-penetrant individuals (II-5 and I-3), although (I-3) had an anxiety disorder diagnosis and is an obligate carrier of any pathogenic variant in this pedigree in an autosomal dominant model. Linkage analysis for a psychiatric phenotype and the identified variant showed an LOD score of 2.2, the family’s maximum for a penetrance of 0.80.

The variant in *CASKIN1* produced an amino acid change (*CASKIN1 D1204N*) and showed significant evidence of functionality based on bioinformatic analyses (Appendix A). The variant was completely absent from all databases and *CASKIN1* has a gnomAD missense Z = 1.79 and pLI = 1 score, indicating its intolerance to variation and pointing to the potential functionality of *CASKIN1 D1204N*. *CASKIN1* is highly conserved and has six N-terminal ankyrin repeats, an SH3 domain, two SAM domains, a proline-rich region, and a C-terminal domain (Figure 2A). *D1204N* resides in the proline-rich region of *CASKIN1*. Although this region’s function is largely unknown, it has been shown to be an intrinsically disordered region and binds Abi2 among other proteins in vitro, providing evidence of its biochemical function [41]. Abi2 is involved in processes linked to cell growth and differentiation [42].

### 3.3. Low SCZ PRS in Non-Penetrant Individuals for CASKIN1 D1204N

Penetrance of a risk allele may be dependent on many factors, such as environmental exposures and genetic background. While we do not have the data to explore the former, we could explore whether the penetrance of the *CASKIN1* variant may be related to the overall genetic risk of the individual as reflected in PRS. We used the software PRSice version 1 [30] and the PGC2 (pgc.unc.edu) results as a reference to calculate PRS for all individuals that carried the variant (Appendix A) and see whether it differentiated the non-penetrant individuals (II-5 and 1–3). PRS perfectly predicted non penetrant individuals (r2 = 1), who had lower PRS compared to affected individuals at *p*-value thresholds between 0.1 and 0.0015. Since PRSice [30] calculates PRS at multiple thresholds, we recognized that the statistical significance of this result is difficult to assess, so we calculated PRS for all possible permutations of phenotypes given six penetrant and two non-penetrant individuals. The same result was observed ~1 in 5 times. Although this result is not statistically significant and does not add significant support to the causality of *D1204N* or *CASKIN1*, it is interesting, as it is consistent with PRSs, which are calculated from common low-effect variants, being important for the penetrance of larger effect variants. Considering our proband was diagnosed with BPD, we also performed PRS analysis using the Mullis et at PGC3 BPD GWAS as reference [43]. Higher BPD PRS did not predict non-penetrant individuals at any *p*-value threshold nor differentiate whether a subject has BPD or SCZ. This reflects the fact that the family had more SCZ than BPD diagnosis.

### 3.4. Generating Isogenic iPSC-Derived Glutamatergic Excitatory Neurons Using CRISPR/Cas9

To determine whether the *CASKIN1 D1204N* variant is likely to have functional consequences, we used CRISPR/Cas9 to introduce it into iPS cells (Figure 2C–F and Figure 3). Because generating a single point mutation using CRISPR/Cas9 through homology directed repair (HDR) has been shown to be inefficient, especially in stem cells [44], we introduced the variant along with three additional changes in *CASKIN1* (Figure 2C). These additional changes abolished the PAM site and the gRNA binding site and removed a restriction enzyme binding site to facilitate screening (Figure 2C). While we initially planned for *D1204N* to be the sole coding change in the region with other changes being synonymous, a design oversite led to three non-synonymous changes (A1206S, H1207N, and P1210S) (Figure 2C). Of these, two have been reported in gnomAD as benign (https://gnomad.broadinstitute.org/variant/16-2229753-C-A?dataset=gnomad_r2_1, https://gnomad.broadinstitute.org/variant/16-2229741-G-A?dataset=gnomad_r2_1) and the third has never been reported. All three variants are within six amino acids of *D1204N*. We considered that, while falling short of unequivocally proving the functionality of the specific variant, this is still a good model to test the significance of the mutated region, so we proceeded with analysis. This is important as it is not known whether this region is of functional significance in humans. In total we derived six edited cell lines carrying mutations on and near *D1204N*, and six unedited lines that underwent the same editing process but did not carry any mutations. (Figure 2D–F). All lines were free of off-target editing at all identified candidate genomic regions (Appendix A).

We differentiated these lines into excitatory neurons using NGN2 induction as previously described [45] (Figure 3, also see methods). The derived neurons expressed multiple neuronal and glutamatergic markers, as we have previously reported for this protocol [46] and as we saw after RNA sequencing (Appendix A).

### 3.5. Changes in the Proline-Rich Region of CASKIN1 Cause Significant Transcriptomic Changes

After initial data processing (see methods), principal component analysis (PCA) revealed two outliers along the first principal component (PC1) (Appendix A). The outliers were one unedited and one edited clone, suggesting a potential technical issue with these samples. This leads us to remove them from further analysis. We found a total of 368 differentially expressed (DE) genes (at adj. *p* < 0.05), of which 232 were upregulated and 136 were downregulated in the edited clones compared to the unedited clones (Figure 4A, Appendix A). The relatively large number of DE genes alone suggests that the edits introduced in the proline-rich region of *CASKIN1* at and near *D1204N* have important functional consequences. We performed a Gene Set Enrichment Analysis (GSEA) [47,48] across 36 manually curated gene sets of canonical pathways related to the nervous system and glutamate signaling and metabolism (Appendix A) showed that 13/36 of these sets were enriched for genes up regulated in the edited clones. No enrichment was observed in the down-regulated genes. We then used the STRING database of protein–protein interactions (PPI; string-db.org) to test whether the DE genes make a meaningful set of proteins with related functions. We found a strong enrichment in PPI with 89 observed edges as compared to the expected 47 for a random set of genes (*p*-value: 2.5 × 10^−8^) (Appendix A). STRING also reported significant enrichment of our DE genes in the following functional categories: regulation of synaptic vesicle exocytosis; chemical synaptic transmission; postsynaptic; regulation of neurotransmitter secretion; modulation of chemical synaptic transmission; cell morphogenesis involved in differentiation; neuron differentiation; generation of neurons; nervous system development; neurogenesis. The obvious relevance of these categories in brain function and development provides a strong link between mutation in this domain of the gene and psychiatric disease.

### 3.6. Electrical Activity Differences in Edited Cells during Maturation

We used live confocal Ca^2+^ imaging to assess the activity of the reference and edited neurons and potential differences between them. We quantified Ca^2+^ activity at three time points, day in vitro (DIV) 21, DIV 28, and DIV 35 for three edited and three unedited iPSC-derived neurons originating from the same cell line. All neurons were electrically active and showed spontaneous and synchronous network activity (Figure 4B and Appendix A). We observed no difference between the edited and unedited clones in the number of active neurons (Appendix A), proportion of active neurons (Appendix A), or synchronicity of events (Appendix A). At DIV 21 spontaneous activity was similar for both genotypes (Figure 4B). At DIV 28, however, it increased and became higher in the edited cells (Figure 4B). DIV 35 activity also increased for the unedited neurons while remaining unchanged for the edited neurons, the two now showing no significant differences (Figure 4B). It appears therefore that while there are no observable differences at DIV 35 activity, the speed at which the cell lines reach the same level of activity differs. This type of difference can be of great significance if present during the highly orchestrated process of central nervous system development. We note that the statistical significance we see on DIV 28 does not withstand multiple correction testing and can only be viewed as a result that warrants follow up.

## 4. Discussion

In this article, we report on a pedigree of a three-generation family that segregates psychosis in an autosomal dominant fashion with a penetrance of ~80% based on inheritance and linkage data. We identified a single coding variant (*CASKIN1 D1204N*) that segregates with the disease with a frequency allowing it to fit the observed inheritance. *CASKIN1* is expressed in the brain and its coding protein product has been shown by others to interact with the coding protein product of *NRXN1*, a known SCZ-associated gene, at the pre-synapse [20,49,50]. *D1204N* resides in *CASKIN1*’s proline-rich region, which has been shown to bind to ABI2 in vitro, indicating that it is biochemically functional. ABI2 is a protein involved in growth and differentiation [41,42]. The functional role of *CAKSIN1*’s proline-rich region is supported by our analysis of the iPSC-derived glutamatergic neurons after CRISPR editing of the region. Further, the gene expression changes observed following the introduction of variants in *CASKIN1*’s proline-rich region by genome editing is consistent with the known functions of *NRXN1* and *CASKIN1*. This supports the validity of our results and implicates *CASKIN1* in the development of psychosis. Our observation that family members without psychosis who carry *CASKIN1 D1204N* have the lowest PRS for SCZ in the family at certain *p*-value thresholds is interesting, despite not reaching statistical significance due to the very small sample size. Through permutation testing, it is suggested that for the given family structure this would only happen one in five times. Although a relatively weak result, it is in agreement with the idea that the penetrance in Mendelian inheritance may be impacted by genomic effects [51] and supports a polygenic threshold model for SCZ [52,53]. Additional research investigating the effects of PRS when Mendelian inheritance is observed in complex phenotypes will eventually answer this question.

Our work is not the first to suggest a link between *CASKIN1* and psychiatric phenotypes. Katano et al. generated *CASKIN1* knock out (KO) mice and showed that they have enhanced nociception and anxiety-like behaviors compared to their wild-type littermates. They also observed that the KO mice had low memory retention in the Barnes Maze test, and interestingly strong freezing responses in contextual and cued-fear conditioning tests with or without a cue tone [54]. Weng et al. have shown that in Drosophila Caskin, which shares homology with vertebrate Caskin, is a neuronal adaptor protein required for axon growth and guidance [55]. This aligns with our observed transcriptome analysis results.

The results from our study are in line with prior research studying the functional effects of the genes implicated in the *NRXN1* pre-synaptic signaling pathway. The increased expression of genes related to glutamatergic transcription in our edited cells identified by GSEA aligns with a study of CASK KO in iPSC-derived neurons [21]. In this study, the authors differentiated CASK KO iPSCs into excitatory glutamatergic neurons using the same NGN2 protocol we used, performed RNA-seq on KO and control neurons on day 7 and observed an upregulation of gene networks related to neuronal cell adhesion, neurite outgrowth, and cytoskeletal [21]. Our results on *CASKIN1* on day 21 are in line with their day 7 results, but an identical timepoint is missing [21]. In the same study, they used MEA to measure neuronal network activity on days 21 and 28 and observed a decrease in the firing rate of neuronal spikes on both days in the CASK KO cells. Although this contradicts what we see in the frequency of calcium transients of the *CAKSIN1* edited cells, it is important to note that the approach used in their study differs from ours and therefore could explain the difference in their results. In another example, *NRXN1*α +/− neurons derived from autism spectrum disorder patients’ iPSCs had a significantly higher frequency of spontaneous Ca^2+^ transients compared to controls [56]. Although the measured neurons were day 100 cortical excitatory neurons using a directed differentiation protocol, they align with the quicker increase in action potential frequency we see in our edited cells during maturation, as observed in the Ca^2+^ imaging experiments. Neurodevelopment is a highly orchestrated process with many genes and cellular processes changing in concert over time, and the artificial in vitro neuronal differentiation protocols used by most investigators, including ourselves, certainly do not faithfully recapitulate the in vivo process. However, it is clear that any disruption in this process can be detrimental and lead to significant neurodevelopmental deficits.

Our transcriptomic analysis provides a significant validation of our approach to investigating the functionality of the proline-rich region of *CASKIN1* at and near our variant of interest. *CASKIN1* has been shown by many to be important for axon growth, guidance and neuronal development [21,55,57,58]. Definite proof of *CASKIN1*’s involvement in psychosis can only come from identification of additional families with similar pedigrees and mode of inheritance, yet our study strongly supports that *CASKIN1* and the pre-synaptic signaling pathway involving it [3,21,22,23] and *NRXN1* warrants intensive study.

It is important to note some limitations when interpreting the results of our study. First, exome sequencing could have missed rare non-coding variants of large effect or large insertions and deletion that might be responsible for the disease in this family. While it is difficult to rule this out, even with whole-genome sequencing, it does not take away from the array of consistent results we observe. Second, a small mosaic gain of ~800 kb (see QC in methods) was observed in some clones and could have influenced the results from the in vitro neurons. Given the mosaic nature of the gain, and that the only four genes in this region are already known to be sometimes duplicated without phenotypic consequences (e.g., essv11597514 and nssv3639434), along with the fact the four genes had no significant expression differences between our edited and unedited cells, we consider it very unlikely to be a significant confounder. The agreement of our transcriptome results with what is expected for the specific gene we modified rather than a random genomic change further supports this conclusion. Finally, we recognize that we fall short of proving the functionality of the specific variant we identified in the family (*CAKSIN1 D1204N*) as nearby changes were also introduced. We do, however, show the functionality of the specific protein domain in a narrow region including our variant. The exact mechanism that the introduced changes contribute to cannot be assessed from our data and remains an important subject for future interest. For these reasons, we are not making claims on *CASKIN1* and the specific variant causing psychosis. Nevertheless, we feel that the genes’ known functions, the segregation in the pedigree along with the functional evidence we present here, support this gene as an excellent novel candidate involved in psychiatric disease risk, warranting extensive further examination of *CASKIN1* and the *NRXN1* pre-synaptic signaling pathway.

## Figures and Tables

**Figure 1 genes-14-00177-f001:**
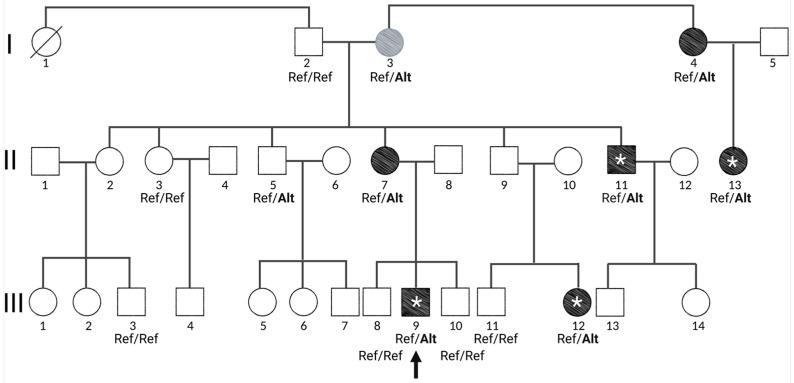
Three-generation family presenting with autosomal dominant inheritance of psychosis. Generations are marked in Latin numerals, I, II, III. Proband is marked with an arrow. Females are circles and males are squares. Individuals with a diagnosis of psychosis are shaded with black and individual I-3 who has anxiety disorder is shaded in gray. Exome sequencing was performed on individuals that are marked with an asterisk (*). Genotypes of the *CASKIN1* variant for all individuals where DNA was available are indicated under the individual’s number where Ref = reference allele and Alt = alternative allele. Only individuals with a genotype were clinically examined.

**Figure 2 genes-14-00177-f002:**
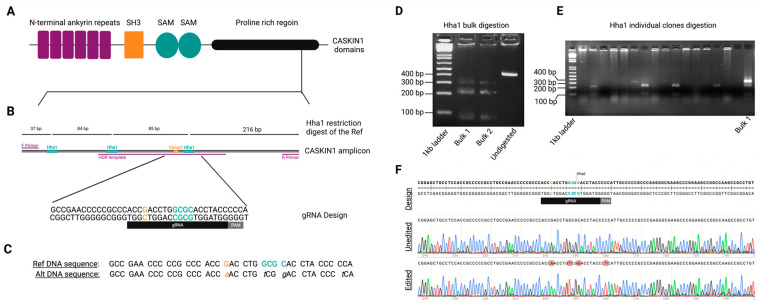
Successful editing of the proline-rich region of *CASKIN1*. (**A**) *CASKIN1* protein domains. *CASKIN1* D1204N is present in the C-terminal proline-rich region. (**B**) *CASKIN1* amplicon depicts the PCR primers (purple) designed to capture the variant (orange) and 3 Hha1 (teal) sites. gRNA design shows the sgRNA (black) and PAM (grey) used to generate edits in the region. Top line captures the Hha1 restriction digest results for an unedited reference line. (**C**) Cell line DNA sequence before and after CRISPR editing. Editing abolishes Hha1 site. (**D**) Screen of a PCR followed by Hha1 restriction digest of two transfected wells. Successful editing is shown as two restriction digestion patterns, reflecting two populations of cells, edited and unedited. (**E**) Screen of clones derived from single-cell cloning. Lane 3 shows a mixed population, lane 7 shows an edited clone and lane 11 shows an unedited clone. (**F**) Example sanger sequencing chromatogram of the cell line before and after editing.

**Figure 3 genes-14-00177-f003:**
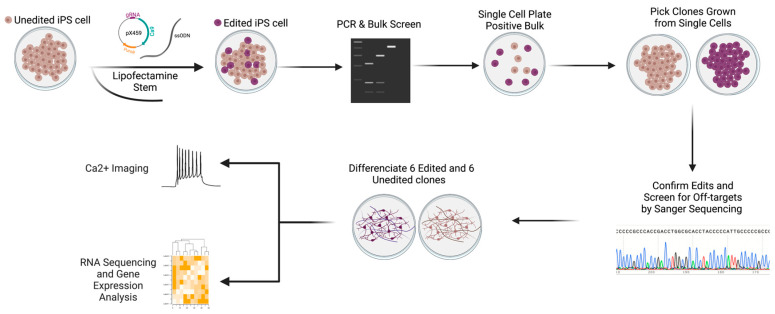
Experimental design and flow of CRISPR editing. Unedited iPS cells were transfected by lipofection with pX459 plasmid, which included a clone in sgRNA, Cas9, and a puromycin resistance gene along with an ssODN containing the changes. Bulk populations of cells were screened by PCR and restriction digestion to look for indications of successful editing. Wells with mixed populations of edited and unedited cells were sparsely plated into single cells. Clones grown from single cells were screened for the edit by Sanger sequencing. Six edited and six unedited clones were chosen and screened for off target edits. The same clones were differentiated into glutamatergic neurons using NGN2. Edited and unedited cells were compared by calcium imaging and RNA sequencing.

**Figure 4 genes-14-00177-f004:**
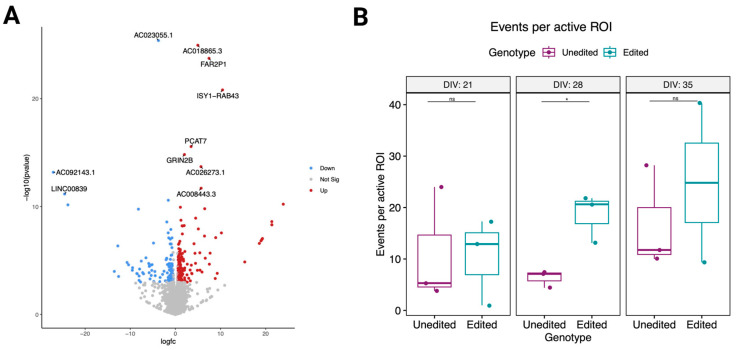
Introducing variants in the proline-rich region of *CASKIN1* causes significant transcriptomic and neuronal activity changes in glutamatergic iPSC-derived neurons. (**A**) Volcano plot showing RNA-seq significantly DEG (padj < 0.05) in color plotted against their magnitude of change (log fold change) where all dots in blue are downregulated and all red dots are upregulated. Each dot is a gene and the top 10 genes are labeled. (**B**) Calcium imaging data showing the number of events per active ROIs in edited (teal) and unedited (purple) cells across 3 time points: DIV21, DIV28, and DIV35, where the edited cells are seen to be more active on DIV28 (nominal *p*-value: 0.0348) compared to DIV21 (*p*-value: 0.941) and DIV35 (*p*-value: 0.692).

## Data Availability

Transcriptomic data will be made available in the Gene expression Omnibus database upon acceptance of the manuscript.

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
