# Peer review of "A Missense Variant in CASKIN1’s Proline-Rich Region Segregates with Psychosis in a Three-Generation Family"

_genes, 2023, doi:10.3390/genes14010177_

Round 1

Reviewer 1 Report

Walsh et al. identified a missense CASKIN1 variant that segreaged with psychosis in a Greek pedigree and examined functional effects of CASKIN1 variants using iPSC derived glutamatergic neurons. This is an interesting paper. However, several issues should be addressed before considering publication.

1.       The title and abstract were misleading. In the title, the NRXN1 interactor should be removed, and disrupts a functional domain goes too far. In the abstract, the description of CASKIN1 should be removed, and the results of calcium imaging should be added.

2.       Line 45, moderate should be small.

3.       Line 65, the authors should describe what proteins do CASK, LIN7A, CASKIN1, and APBA1 genes code.

4.       Line 302 and Table S1, did the proband have bipolar I or II?

5.       Table S1, I-14 should be I-3. This individual was diagnosed as healthy using MINI and DSM-IV but had anxiety disorder. Therefore, diagnoses made using MINI and DSM-IV were not reliable. Fig. 1, what did squares and circles mean?

6.       Line 325, 3 should be 4.

7.       Table S2, the authors should provide the chromosome position and allele of variants identified.

8.       The authors should provide PRS of bipolar disorder for each individual because III-9 and III-12 had bipolar disorder and schizoaffective disorder bipolar type, respectively. They should also provide the levels of PRS of schizophrenia and bipolar disorder in participants compared to patients with these disorders and controls in the previous studies.

9.       Fig 4 and Fig S4, corrected p-values should be provided.

10.   Line 417, the authors claimed the disruption of CASKIN1’s proline-rich region by genome editing. However, they did not show evidence for the disruption of CASKIN1’s proline-rich region.

Author Response

Reviewer 1:

  • In response to comment 1 which suggested that the title was going too far in its statements. We have revised the title to tone it down. It is now “A missense variant in the CASKIN1 proline-rich domain segregates with psychosis in a three-generation family”
  • In response to comment 2, we have changed “moderate” to “small” as suggested.
  • In response to comment 3 we made clearer the short description of the functions and relationships of the CASK, LIN7A, CASKIN1, and APBA1 gene products in our introduction.
  • In response to comment 4 we have added in the diagnosis that the proband had bipolar I.
  • In response to comment 5 we apologize for the lack of clarity. In the specific fields of the table we should not have written “healthy” as this is inaccurate, rather it should be “no psychotic disorder”. MINI and DSM-IV did diagnose the anxiety disorder as noted in the next column. We have corrected the table to clarify.
  • In response to comment 6 the reviewer is correct, and we have corrected 3 to 4, thank you for noticing this typographical error.
  • In response to comment 7 we have now added the chromosome position and allele of variants identified.
  • In response to comment 8 we have performed now PRS analysis for BPD and did not find it to contribute to the penetrance of the variant. We now added this result. The question we asked with the PRS analysis was about the penetrance of the identified variant so only affected individuals were included. We have now added a table (S3) with all PRS for affected individuals for both SZ and BP. As for adding individuals from previous studies, to be meaningful this would require primary genotype data for running them together with ours, which we do not currently have in hand.
  • In response to comment 9 we have added the exact nominal p-values so correction can be calculated. Bonferroni correction is not customary for these types of analyses, where the purpose is exploratory, and comparisons are relatively few and with an unknown correlation structure. Nevertheless, it is correct that if so adjusted, our p-values should not be considered study-wide significant, so we have added this limitation in the results.
  • In response to comment 10 we apologize if our wording was not accurate. What we mean is that we modified the protein sequence, not necessarily disrupted the proline-rich region. As a result of this modification, we observed many significant transcriptomic changes which we believe is evidence of a functional consequence, but we agree it does not necessarily show disruption. We have now changed the wording and clarified this in the text.

Reviewer 2 Report

In this manuscript Wahbeh MH et al, concluded that CASKIN1 is an excellent candidate gene for psychosis development with high penetrance in by showing segregation of this D1204N variant in three-generation of a large family with SCZ and bipolar and identified significant cellular changes in iPSC derived neurons, including transcriptomic changes in genes involved in neuronal differentiation and nervous system development, as well as changes in the frequency of action potentials during differentiation. Overall, the manuscript is well written, and figures are well presented, however, most of the conclusions are primarily based on in silico predictions, analysis and findings. 

1)      Figure 1: phenotype description of II-5 male carrier and rational should be provided for possible low penetrance, variable expression (I-3) or environmental factors resulting in the normal or milder phenotype

2)      Authors observed transcriptomic upregulation and downregulation of gene networks related to neuronal cell adhesion and function as well as observed possible protein-protein interaction in the in silico STRING analysis, however, no interaction was confirmed from further in vitro/vivo experiments.

3)      Additionally, authors did not provide any mechanistic rational to connect this CASKIN1 variant with observed transcriptomic changes of neuronal genes. This is important to understand and disease pathogenesis and mechanistic role of CASKIN1 in the development of phycological phenotype.

4)   To establish connection of this variant with NRXN1 pathway, an in vitro/vivo experiment should be included in the study to show specifically the effect of this CASKIN1 D1204N variant on NRXN1 expression or CASKIN1-NRXN1 interaction to provide mechanistic connection of this variant with NRXN1 and overall psychosis phenotype 

Author Response

Reviewer 2:

In the general comments it is mentioned that most of our conclusions are based on in silico predictions. We would like to point out that the data in the STRING protein interaction database come mainly from publications and curated databases that have experimentally linked the proteins together in a variety of approaches and indicate functional and physical associations that are not just in silico predictions. All protein association evidence can be accessed through STRING. The same is true for gene sets in the GSEA libraries and for all the papers we site linking CASKIN1 to NRXN1 intracellular signaling. It is not clear to us which part of our analysis is based on in silico predictions

  • In response to comment 1 individual II-5 was found psychiatrically healthy by the MINI and DSM-IV as reported on our diagnosis table. There are no further phenotypes to report. The rationale for variable expressivity and incomplete penetrance comes directly from the data. These individuals are carriers but one has a significantly different psychiatric phenotype and the other no psychiatric phenotype, which are the definition of variable expressivity and incomplete penetrance. The mention of environmental factors is simply a discussion of possible reasons, along with the polygenic risk score. We have explored schizophrenia PRS as a potential explanation comparing the penetrant and non-penetrant individuals suggesting that the non-penetrance may be due to their overall more protective genetic makeup. We have made edits to the text when needed to clarify.
  • In response to comment 2, as we mentioned above, these interactions have been shown by others and are derived from published data, which is why they are reported in STRING. It is beyond the scope of this research to duplicate these results and it would be extremely time consuming to do so.
  • In response to comment 3, it is true that we did not provide a specific mechanistic rational to connect this variant with observed transcriptomic changes of neuronal genes, this however would be pure speculation. We report that changing the sequence of this region of CASKIN1 leads to significant transcriptomic changes, suggesting some functional consequence. The exact mechanism leading to these changes is within our future directions but would be a lengthy research project, far beyond the scope of this paper. We have added a few words about this in the end of our discussion. Here we simply report data showing that CASKIN1 should be considered as a strong candidate for psychiatric disease
  • Comment 4 mentions that an in vitro/vivo experiment should be included in the study to show specifically the effect of this CASKIN1 D1204N variant on NRXN1 expression or CASKIN1-NRXN1 interaction. Regarding the expression, our data already shows that there is no such effect. As for showing effects of the interaction, this is no simple task because, as we mention, the interactions are tripartite and CASKIN appears to compete with MINT1 to form complexes with CASK and VELIS, which complexes in turn interact with NRXN1. Resolving these complex interactions in vitro is difficult and likely unreliable. Nevertheless, our lab is following up this finding with similar mechanistic analyses. Because however these are time consuming, we feel that, while they are important follow up work, they are beyond the score of this manuscript which aims to report our current results that suggest CASKIN1 is strong candidate for psychiatric disease.

Round 2

Reviewer 1 Report

Fig. 1, what did squares and circles mean?

Author Response

Thank you for the opportunity to revise our paper. In response to the reviewer comment we added a sentence in the legend of figure one stating that squares are males and circles females. We also took this opportunity to correct a typo in Table S1 (individual with ID i-14 should be I-3, this has now been corrected) and to sort the Table by ID so that it is easier to read.